# Development and validation of a questionnaire, the REST-Q Fire, to identify teamwork-related resources and stressors in firefighting operations

Lena Heinemann[1]*, Fabienne Aust[2], Corinna Peifer[2], Maik Holtz[3], Christian Miller[3], Vera Hagemann[1]

**1** Business Psychology and Human Resources, University of Bremen, Bremen, Germany, **2** Department of Psychology, University of Lübeck, Lübeck, Germany, **3** Cologne Fire Department, Institute for Security Science and Rescue Technology (ISR), Cologne, Germany

* lena.heinemann@uni-bremen.de

**Data Availability Statement:** All relevant data for this study are publicly available from the OSF

## Abstract

In the complex work environments of firefighting teams, it is often human error or difficulties in teamwork that lead to dangerous situations. To prevent these, it is essential to know the teamwork-related stressors and resources in firefighting operations. A measurement tool is needed to evaluate these stressors and resources. A successive instrument development process was conducted to identify the relevant teamwork-related stressors and resources in firefighting operations. First, interviews with experienced firefighters, and second, a document analysis were conducted and evaluated to provide an overview of the teamwork-related stressors and resources. Thereupon, a questionnaire, the REST-Q Fire, was developed asking about the experienced frequency and intensity of the identified teamwork-related stressors and resources in firefighting operations. Afterwards, an online study with firefighters was conducted ($N = 491$). CFAs confirmed the assumed structure of the REST-Q Fire and a positive correlation of the intensity of stressors with stress was shown ($r = .19 - .27$). Further, the resources were overall more frequently and intensively perceived than the stressors. The most important stressors were 'behavior of leaders' ($M (SD)_{frequency}$ = 2.80 (0.83), $M (SD)_{intensity}$ = 3.59 (1.12)), and 'behavior of team members' ($M (SD)_{frequency}$ = 2.77 (0.75), $M (SD)_{intensity}$ = 3.59 (1.05)). The most important resources, on the other hand, were 'knowledge about skills and behavior of team members' ($M (SD)_{frequency}$ = 3.96 (0.63), $M (SD)_{intensity}$ = 4.24 (0.78)), 'watch out for/ check on each other' ($M (SD)_{frequency}$ = 3.96 (0.70), $M (SD)_{intensity}$ = 4.20 (0.80)), and 'reliability of team members' ($M (SD)_{frequency}$ = 3.96 (0.51), $M (SD)_{intensity}$ = 4.16 (0.73)). As a result, training needs for trainees in the fire service and experienced firefighters were derived.

repository (https://doi.org/10.17605/OSF.IO/DFHPQ).

**Funding:** This work was supported by the German Social Accident Insurance (DGUV) [grant number FP-433]. The funders had no role in study design, data collection and analysis, decision to publish, or preparation of the manuscript.

**Competing interests:** The authors have declared that no competing interests exist.

# Introduction

The work of firefighters is vital to society, as we all depend on emergency services to help us in potential emergencies. Firefighters must deal with dangerous situations and experiencing stress is part of their daily work [1]. Not only must they always work together as a team but they also depend on well-functioning team processes. Poorly running team processes on the other hand can increase the risk of accidents and might have far-reaching consequences [2, 3] as statistics from the Firefighters' Accident Insurance Fund (FUK) for Germany show. Accidents repeatedly occur during fire service activities, especially during firefighting. From 2019 to 2021, firefighting accounted for between 19% and 39% of all reportable accidents [4]. The same figure applies to firefighters in the USA [5] where eighteen firefighters died in firefighting operations in 2019 [6]. Therefore, identifying teamwork-related stressors and resources to counteract the stressors or promote the resources is important for safe and successful work [2, 3].

In 2005, two firefighters lost their lives in a devastating fire in Tübingen. During the firefighting operation, the attack team entered the attic of a burning building without passing this information. As the operation progressed, the attack team's oxygen supply was exhausted, resulting in a mayday situation. The support team that advanced after the mayday situation was declared came too late. One of the reasons for this was that the team lacked information about the exact position of the attack team. In this firefighting operation, the attack team did not communicate according to the instructions. As a result, the rest of the team could not understand how they were proceeding and where they were in the building [7].

In another case during a firefighting operation in the USA in 2016, the first attack team proceeded to locate, contain, and extinguish a spreading fire in a residential building. At the same time, a second attack team laid an attack line to the other side of the building and noticed a large amount of fire and smoke coming from inside the building. The incident commander decided that the second attack team should discharge water into the building from their position. Several attempts to reach the first attack team were unsuccessful. Nonetheless, the second attack team began to discharge water without making sure that the first attack team was aware of the plan and was in a safe area. The heat and water vapor eventually forced the first attack team to retreat [8]. Before they released water, the second attack team should have made sure that the first attack team had left the building or was in a safe area. In addition, the incident commander did not ensure that all relevant personnel were aware of the plan before implementing such a major change in tactics.

On the whole, there were no technical difficulties in any of the operations described, but the dangerous development is solely due to human error. The fatal consequences in the first example and the danger that the first attack team was put in in the second example could have been prevented if the teamwork had worked better.

As further evidence in support of this thought, Moore-Merrell et. al [9] identified lack of situational awareness (37%), lack of fitness (29%), and human error (11%) as the most common reasons for firefighter injuries in (general) operations in the USA. That shows that two of the three most common reasons are related to human factors. Various reasons arising from teamwork (decision-making, lack of communication, breach of standards, human error, and lack of situational awareness) together account for 38% of all firefighter injuries during firefighting operations [9]. Another study identified nine stressors that contribute to firefighter fatalities during operations in Spain, of which two are teamwork and decision-making under stress [10].

However, the majority of research on job-related stressors in firefighting operations does not explicitly deal with teamwork. Instead, it uses the following distinctions: organizational stressors and operational stressors [11–14]. Operational stressors are inherent in the job and can be influenced little to not at all (e.g. threatening situations, noise, smoke, fatigue due to

shift work). Organizational stressors arise from the context of the job and can be influenced by the organization (e.g. high workload, lack of staff, insufficient training, insufficient resources). In some studies, organizational stressors also include aspects of teamwork (e.g. little decision-making autonomy [12], role conflicts [12], conflicting tasks [13], inconsistent leadership style [11], lack of support between colleagues [11, 13]). In addition, few studies further subdivide the stressors into e.g. managerial, operational, personal, and interpersonal stressors [15] or explicitly examine psychosocial stressors [16, 17].

As described above, the operational stressors that occur during firefighting operations are already quite well-researched but can hardly be influenced. Teamwork-related stressors are only incidentally considered as a small part of these, if at all. Teamwork skills, however, can be trained and can thus contribute to stress reduction in firefighting operations [18, 19]. It is expected that improved teamwork skills lead to fewer critical situations, which strengthens accident prevention [20–22]. Consequently, it is essential to determine the needs of firefighters more accurately and to identify which teamwork-related stressors and resources are relevant in the field, to enable teamwork to be improved [23, 24]. In addition, resources are often disregarded, although they are important to support successful and stress-reduced firefighting work. Some studies deal with coping mechanisms that can be seen as resources (e.g. [25]), but these also do not explicitly refer to teamwork.

Accordingly, it is necessary to develop appropriate measurement tools that reflect the specific needs to conduct future research regarding teamwork-related stressors in firefighting operations. In addition, teamwork-related resources should be considered. The absence of a stressor does not automatically mean that a resource is present, but it must be explicitly recorded. To better understand which aspects are important for a measurement tool of teamwork-related stressors and resources specific to firefighting operations, it is necessary to understand (1) the specifics of teamwork in so-called High Responsibility Teams (HRTs), in this case especially firefighting teams, as they differ significantly from other teams and (2) the definition and importance of non-technical skills.

## High Responsibility Teams

In healthcare, police work, aviation, and firefighting, the work environment is characterized by a high potential for danger and risk, and the stakeholders must consequently act with utmost reliability under high pressure of time. Accordingly, the organizations in these areas are referred to as High Risk or Reliability Organizations (HROs) [26]. HRTs operate in these organizations and face many particular challenges in their day-to-day work [27]. For example, they are often under pressure from the public or the media during their work, they bear responsibility for their own lives and the lives of others, and moreover, it is typically impossible for them to interrupt their work or adjust their actions afterward. HRTs also often work in changing compositions of teams and with people they do not know. Thus, in addition to physical stress, they are often exposed to psychological stress [28]. Despite these dynamic conditions in which teams must act quickly and confidently, smoothly functioning teamwork processes are particularly important for successful collaboration [29, 30]. High reliability cannot be reached when team members do not coordinate their activities [31] and a mistake or a misunderstanding can cause substantial damage [32]. For healthcare, it could be shown that teamwork is one of the three leading contributing factors to problems [33] and that critical situations often occur due to non-technical problems, such as failure in communication, and not necessarily due to technical difficulties [34].

In the context of firefighting, no research has been found that systematically surveyed teamwork-related stressors and resources. It is essential to take the special working conditions of

these HRTs into account while developing a measurement tool to identify teamwork-related stressors and resources. Furthermore, the knowledge of non-technical skills is vital for successful teamwork and should be considered during development.

## Non-technical skills

For the fulfillment of firefighting tasks physical skills as well as technical/tactical knowledge (e.g. for the correct laying of hoses) are required. In addition, the team members need non-technical skills. These are cognitive, social, and personal skills to work together in the best possible way, achieve goals, and make as few mistakes as possible [24, 35, 36]. Effective teamwork is supported by non-technical skills like communication, coordination, decision-making, leadership, and the development of a shared mental model [2, 3, 37, 38]. These five non-technical skills interact to form the basis for relevant stressors and resources in the context of teamwork and are presented below.

*Communication* is a prerequisite for interaction between team members and is therefore essential for teamwork. It is particularly important in complex situations (such as firefighting) when stress can cause team members to focus on their individual tasks and hinder team interaction. However, it is particularly important in these situations that information is given to the right person at the right time. When this is omitted, coordination is no longer possible as there is no shared knowledge about the current status and tasks of the individual team members [38, 39]. As a result, the lack of communication can lead to unsafe situations [2].

*Coordination* makes it possible to manage the interdependencies between the roles and tasks of team members as well as possible conflicts between their goals [2] without wasting valuable resources [40]. If coordination breakdowns occur, they can lead to a loss of time or errors, such as the omission of work steps. Without correction, this can lead to further errors by other team members and/or critical incidents [39].

*Decision-making* is the process of evaluating a particular situation to then choose an appropriate option. This is usually done in a continuous process of reviewing the circumstances to adjust the chosen option if necessary. Decision-making skills are particularly important in HRTs, where decisions often have to be made under time pressure and stress [41]. Psychological factors (such as impaired memory caused by exhaustion in complex work contexts) negatively influence decision-making behavior [3].

*Leadership* is responsible, among other things, for supporting team problem-solving through e.g. coordination processes, cognitive processes (such as building a shared mental model), an appropriate team climate and motivation, as well as assigning tasks and assessing team performance (for an overview see [38]). The team leader guides the team to successfully perform their tasks while considering the safety standards [41].

*Situational awareness* was identified by Heydari et al. [42] as the most important cognitive factor influencing the performance of industrial firefighters. In particular, shared situational awareness is significant to enable team members to adapt to a (rapidly) changing environment [39]. Moreover, it is only possible to refer to a shared mental model, central to successfully conducting firefighting operations, with shared situational awareness (e.g. [38]). A shared mental model means that the team members are clear about the common goal and have a shared understanding of what tasks need to be done to achieve this goal and who is responsible for what. Furthermore, they have a shared understanding of the knowledge and skills of each team member [41, 43].

Given these points, knowledge about relevant teamwork-related stressors and resources during firefighting operations is essential to identify problems with non-technical skills in that context. On the one hand, this is important to be able to develop and evaluate targeted training

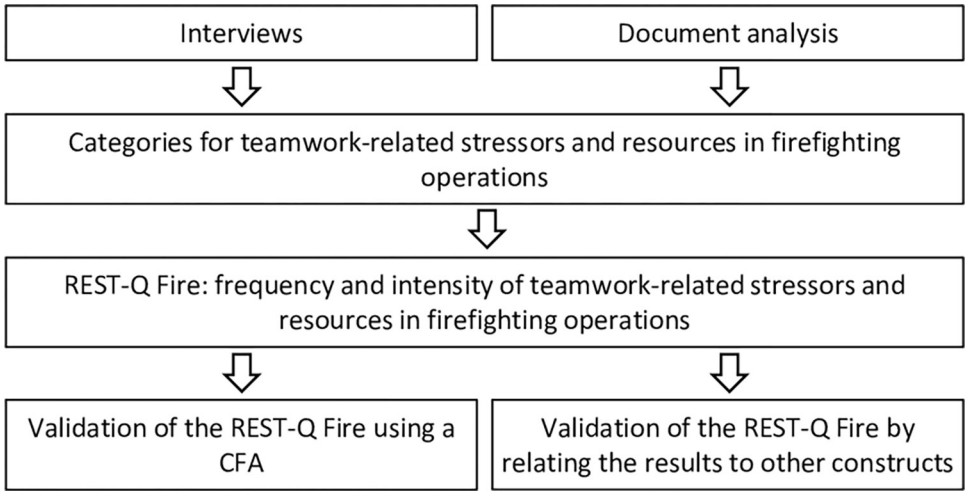

**Fig 1. Study procedure.**

measures for trainees in the fire service and, on the other hand, to be able to identify and specifically address training needs in existing teams. However, there is yet no instrument to measure this in the context of the fire service. Therefore, this study aims to develop a questionnaire that captures the teamwork-related stressors and resources experienced during firefighting operations. Fig 1 shows all aspects necessary for the process of the present paper.

The two steps of developing and applying the questionnaire are explained below.

## Step 1: Item development based on interviews and document analysis

A qualitative approach was used to learn about teamwork-related stressors and resources in firefighting operations. Interviews with experienced firefighters and a document analysis were conducted and evaluated for this purpose.

### Semi-structured expert interviews and document analysis

**Sampling strategy and description of the sample.** Semi-structured expert interviews were conducted between May 28, 2020 and August 25, 2020 with experienced professional, volunteer, and industrial firefighters. Experienced was defined as at least two years of professional experience or at least five firefighting operations. A contact in a professional fire department was used to reach professional firefighters willing to be interview partners. This contact also resulted in some connections to volunteer firefighters. Furthermore, volunteer fire departments were contacted through private networks. Outreach to industrial fire departments was established through cooperation partners. The study was conducted according to the guidelines of the Declaration of Helsinki and was approved by the Institutional Ethics Committee of the Faculty of Psychology, Ruhr-University Bochum (no. 482, approval issued on: Aug. 3, 2018). Written informed consent was obtained from all subjects involved in the study. The interview duration was announced as 60–90 minutes and the participants could decide for themselves whether they wanted to participate in person or by telephone.

A total of 27 (including 3 female) firefighters participated in the interviews. Eight were primarily or exclusively professional firefighters, 9 were industrial firefighters, and 10 were volunteer firefighters. Just over half (9) of the professional and industrial firefighters were equally

active in the volunteer fire department. Participants ranged in age from 22 to 60, with the majority of participants being 26–30 or 31–35 years old (22.2% each). This was followed by the 36–40 and 46–50 age groups (14.8% each). The work experience of the professional firefighters ranged from 3 to 28 years with a mean of 12.00 years ($SD$ = 10.25). Among the industrial firefighters, work experience ranged from 6 to 35 years with a mean of 13.67 years ($SD$ = 10.16). Volunteer firefighters had between 6 and 33 years of work experience with a mean of 19.33 years ($SD$ = 8.43). Participants recalled an average of 102 firefighting operations, in which they were involved as firefighters ($SD$ = 197.96).

**Interview procedure and evaluation.** The interview guide was developed by referring to the preliminary study [1]. At the beginning of the interviews, some demographic information was collected, such as years of work experience and number of experienced firefighting operations. The structure of the interview was based on the Critical Incident Method according to Flanagan [44]. That is, the main part of the interview dealt with a negatively perceived firefighting operation that was examined from multiple facets. To do this, the interviewee was slowly introduced to the firefighting operation situation by being asked specific questions about the situation, such as what time of day it was and what the situation was when approaching the scene. Through the introductory questions, the interviewees should have been able to put themselves back into the situation as far as possible. Subsequently, questions were asked about the teamwork and what went well or badly.

The interviewers kept a summary protocol of what was said during the interview. The advantage of this method is that the interviewees are less intimidated when no recording device is used [45]. This also eliminates the need for transcription after the interviews since a results-oriented interview protocol is already available. Factors such as syntax or other linguistic peculiarities were not considered since a content-only analysis was performed.

Subsequently, the interviews were analyzed using qualitative content analysis based on Mayring [46] to structure the material. By reading and rereading the protocols, teamwork-related stressors and resources in firefighting operations were identified and summarized into subject areas. From this, superordinate categories were formed into a coding framework, into which all segments were gradually classified. In the process, the coding framework was continuously adjusted and subcategories were formed that helped to structure the segments more precisely. The coding was cross-checked by two researchers, and discrepancies were resolved by consensus.

**Document analysis.** For the document analysis, around 115 operation reports were screened for teamwork-related stressors and resources in firefighting. The reports were primarily from the FUK CIRS (a Critical Incident Reporting System for the Firefighters' Accident Insurance Fund), atemschutzunfaelle.eu, and firefighternearmiss.com websites. After reviewing the firefighting operations documented there, a coding framework was created, according to the procedure for the interviews described above, to be able to evaluate the relevant text segments from the documents in a structured manner.

**Results.** The two coding frameworks that emerged from the document analysis and the interviews had a high degree of similarity, so that they could subsequently be merged well, and a common coding framework was developed. As a result, a category system with six superordinate categories and 26 subcategories for stressors as well as 23 subcategories for resources was available (see Table 1).

## Item generation

Items for the REST-Q Fire (**RE**sources and **S**tressors of **T**eamwork in firefighting operations **Q**uestionnaire) were formulated based on the coding framework. The basic idea was to design

**Table 1. Categories for teamwork-related stressors and resources in firefighting operations.**

| Superordinate categories | Subcategories |
|---|---|
| Communication | Passing on information within the squad |
| | Passing on information between the fire brigade's task forces |
| | Quality of information |
| | Amount of information |
| | Communication with third parties |
| | Shared situational awareness |
| Supportive behavior | Watch out for/ check on each other |
| | Behavior of team members |
| | Respond to the needs of others |
| | Supportive behavior/ support not possible 1 |
| | Supportive behavior/ support not possible 2 (stressor only) |
| | Reliability of team members |
| Leadership | Hierarchy/ followership |
| | Structure/ hierarchy |
| | Insufficient consideration of hazards and operational standards |
| | Behavior of leaders |
| Shared mental models | Knowledge about common course of action |
| | Knowledge about skills and behavior of team members |
| | Interpositional knowledge |
| Organization/ coordination | Task allocation |
| | Standards and safety measures |
| | Self-initiative and independence 1 |
| | Self-initiative and independence 2 (stressor only) |
| Decision-making | Decision in case of situation change/ deciding between different alternatives 1 |
| | Decision in case of situation change/ deciding between different alternatives 2 (stressor only) |
| | (Support of) Decision makers |

the REST-Q Fire as realistically as possible. This was achieved by describing concrete situations and including both positive and negative aspects of teamwork in firefighting operations. Accordingly, passages from the interviews and the document analysis that represent typical situations for the corresponding subcategory were used as a basis for the situation descriptions. All firefighting-related non-technical skills identified in the qualitative approach were to be covered. To this end, two items were formed for each subcategory: one for a stressor and one for a resource. When formulating the items, attention was given to a uniform sentence structure so that it would be easy for all participants to understand. The situation descriptions were specific to the firefighting context but still broad enough that similar situations experienced by the participants could also be attributed to them. In this process, the situation descriptions were enriched with concrete examples. The content of the items was checked by a subject matter expert to ensure that the correct terms were used. Subsequently, all items were discussed and finalized within the research group. In total, 49 items were generated. The complete item list with the situation descriptions can be found in S1 Table.

Two scales were used for each stressor and resource; one for the frequency of their occurrence and one for the intensity of the experience. For stressors and resources, it is very important to record both the frequency and the intensity, as these can be very different and have a different message. It is also important to prioritize which stressors pose the greatest risk and which resources can provide the most support, as it is usually not financially and

---

**Initial situation:**
During a firefighting operation, a firefighter had important information (e.g. situation report, briefing of new squads) which he was supposed to pass on to the other firefighters of the fire brigade.
And then the following happened:

| | |
|---|---|
| The **information passed on** was **incorrect, inaccurate**, and/or **incomplete**. (stressor) | The **information passed on** was **clear, unambiguous**, and/or **correct**. (resource) |

**How often** did this occur in your previous experienced firefighting operations?
- Five-point Likert scale from (1) never to (5) always

If it happened, how much did this **stress/help** on average?
- Five-point Likert scale from (1) not at all to (5) very strongly

---

**Fig 2. Example for the presentation of the items in the REST-Q Fire.** Example for communication: quality of information.

organizationally possible to address them all [15]. In order to allow participants to select a scale midpoint, a 5-point scale was selected. For frequency, the question asked was, "How often did this occur in your previous experienced firefighting operations?", which could be answered on a scale of 1 (never) to 5 (always). For intensity, the question asked was how much this situation stressed or helped on average. Answers could be given on a scale from 1 (not at all) to 5 (very strongly). In addition, it was possible to state that the stressor or the resource had never occurred before so the intensity cannot be rated. Fig 2 shows the two items for the subcategory *Quality of information* in the superordinate category *Communication* as an example.

## Step 2: Application of the REST-Q Fire

### Sampling strategy and description of the sample

The questionnaire was advertised using private networks, a mail distribution list of the Cologne professional fire department, one of the largest fire departments in Germany, a newsletter of the German Social Accident Insurance (DGUV), and a German firefighting journal. 1135 participants processed the questionnaire, but 644 of them had to be excluded for the following reasons: (1) discontinuation of participation after the demographic data or the first scale ($n = 380$), (2) too little information on the study variables ($n = 255$), or (3) too little experience in the fire service: less than 5 firefighting operations in combination with less than 2 years of professional experience ($n = 9$). Because of the randomized items, data sets that contained at least two complete superordinate categories (of the six superordinate categories shown in Table 1) could still be analyzed. The final data set thus included 491 participants.

Participants ranged in age from 18 to 70 years ($M = 39.70$ years, $SD = 11.03$) and were 94.5% male (5.5% female). They reported being active as a firefighter for 2 to 49 years ($M = 20.28$ years, $SD = 11.00$) and having experienced between 1 and 7500 firefighting operations ($M = 420$ firefighting operations, $SD = 832.03$, *median* = 150 firefighting operations). The majority of participants were primarily active in volunteer fire departments (65.4%), just under one-third belonged to professional fire departments (29.9%), and the remainder was divided into smaller parts (industrial fire departments 3.9%, company fire departments 0.6%, and federal armed forces fire departments 0.2%).

## Questionnaire procedure

An online questionnaire was generated with the newly developed items as well as questions about stress and conflicting goals in firefighting operations (self-developed items). The order of the categories of stressors and resources was randomized to achieve a balanced number of participants for each item.

The online questionnaire was pretested by 20 participants, who commented on the structure, comprehensibility, and feasibility of the questionnaire. After adjusting the questionnaire based on the pretesters' comments (e.g. a more precise definition of what was meant by teamwork in the questionnaire; a more detailed explanation of the context to which the questions relate), it was online from January 18, 2021 to May 25, 2021. It comprised 26 pages in the EFS survey tool and it took around 40 minutes to complete the questionnaire in full. The study was conducted according to the guidelines of the Declaration of Helsinki and was approved by the Institutional Ethics Committee of the Faculty of Psychology, Ruhr-University Bochum (no. 482, approval issued on: Aug. 3, 2018). Furthermore, written informed consent was obtained from all subjects involved in the study.

## Validation of the REST-Q Fire

Several criteria were considered that can be used as indications for the applicability of the REST-Q Fire. In the first place, data showed that the teamwork-related stressors and resources identified through expert interviews and document analysis are part of the everyday work of firefighters. On average, each participant experienced 22 of the 26 stressors and 23 of the 23 resources during their career as a firefighter at least once. Accordingly, the teamwork-related stressors and resources in the REST-Q Fire are relevant to firefighting.

**Confirmatory factor analyses.** Furthermore, confirmatory factor analyses (CFA) were conducted. They verify if the structure of superordinate categories and subcategories assumed based on the interviews and document analysis could be confirmed by the data. The software program R with the package lavaan was used for the CFAs [47]. Four CFAs were conducted— one each for frequency of stressors, intensity of stressors, frequency of resources, and intensity of resources. Due to the nature of the analysis, only those participants who completed the respective category fully were used. This means for the frequency of stressors $n = 329$, the intensity of stressors $n = 96$, the frequency of resources $n = 397$, and the intensity of resources $n = 357$. The intensity of stressors has a low $n$ because many stressors were not experienced by the participants and therefore the intensity could not be assessed.

All newly developed items on the frequency of stressors fit the model. The model fit indicators for the six-factor model with the second-order common latent variable stressor frequency were satisfactory ($\chi^2 = 469.33$, df = 290, p = .000, $\chi^2$/df = 1.62, TLI = .93, RMSEA = .046, SRMR = .048). The cutoff criteria for fit indexes for this and the three following CFAs are taken from Hu and Bentler [48] and Vandenberg and Lance [49]. All factor loadings were above .40 and significant. The internal consistency of the scale stressor frequency was $\alpha = .92$ and for the subscales between $\alpha = .46$ and $\alpha = .80$ (see Table 2). As Ziegler et al. [50] describe, a low Cronbach's alpha is a common problem with short scales and should not necessarily lead to not using these scales. Since the scales all have content relevance and the model fit indicators were still satisfactory, all scales are left in the model.

Regarding the intensity of stressors, all newly developed items also fit the model. The model fit indicators for the six-factor model with the second-order common latent variable stressor intensity were satisfactory ($\chi^2 = 405.85$, df = 293, p = .000, $\chi^2$/df = 1.70, TLI = .89, RMSEA = .068, SRMR = .065). All factor loadings were above .40 and significant. The scale stressor

**Table 2. Internal consistencies of the teamwork-related stressors and resources and subcategories and their correlations with stress.**

| | | Frequency | | | Intensity | | |
|---|---|---|---|---|---|---|---|
| | | Cronbach's α | Stress | n | Cronbach's α | Stress | n |
| Stressors | | .92 | -.02 | 486 | .95 | .24** | 377 |
| | Communication | .80 | -.06 | 416 | .87 | .25** | 247 |
| | Supportive behavior | .76 | -.09 | 396 | .85 | .24** | 165 |
| | Leadership | .70 | .01 | 432 | .75 | .22** | 307 |
| | Shared mental models | .64 | .03 | 425 | .73 | .19** | 281 |
| | Organization/coordination | .70 | .05 | 435 | .73 | .21** | 314 |
| | Decision-making | .46 | .02 | 438 | .70 | .27** | 294 |
| Resources[1] | | .89 | -.04 | 462 | .93 | .07 | 474 |
| | Communication | .81 | -.04 | 421 | .87 | .08 | 399 |
| | Supportive behavior | .70 | -.00 | 427 | .81 | .07 | 412 |
| | Leadership | .59 | -.04 | 436 | .67 | -.04 | 423 |
| | Shared mental models | .67 | -.05 | 438 | .70 | .07 | 433 |
| | Organization/coordination | .53 | -.09 | 438 | .64 | .09 | 436 |
| | Decision-making | - | - | - | .62 | .10* | 430 |
| | Decision-making Item 1 | - | .06 | 459 | - | - | - |
| | Decision-making Item 2 | - | -.06 | 454 | - | - | - |

** *p* < .01 (two-tailed)

* p < .05 (two-tailed); pairwise deletion was used

[1]frequency without decision-making

intensity (α = .95) as well as all subscales showed a respectable or good internal consistency (α = between .70 and .87) (see Table 2).

However, for the frequency of resources, the two items regarding decision-making had to be excluded from the CFA because of poor fit. Therefore, they are not used for the total score. The model fit indicators for the five-factor model with the second-order common latent variable resource frequency were satisfactory ($\chi^2$ = 253.44, df = 182, p = .000, $\chi^2$/df = 1.39, TLI = .95, RMSEA = .036, SRMR = .047). All factor loadings except for two were above .40 and all factor loadings except for Organization/coordination were significant. Nevertheless, organization/coordination remains in the model since the model fit indicators are satisfactory. Furthermore, the factor is still used for the total score due to its contextual importance. The internal consistency of the scale resource frequency was α = .89 and for the subscales between α = .53 and α = .81 (see Table 2).

All newly developed items on the intensity of resources fit the model. The model fit indicators for the six-factor model with the second-order common latent variable resource intensity were satisfactory ($\chi^2$ = 239.48, df = 221, p = .188, $\chi^2$/df = 1.08, TLI = .99, RMSEA = .018, SRMR = .036). All factor loadings were above .40 and significant. The internal consistency of the scale resource intensity was α = .93 and for the subscales between α = .62 and α = .87 (see Table 2).

Overall, the CFAs confirmed the assumed structure to a large extent. This shows that the total scores of the superordinate categories and subcategories can be used for the data analysis and that the development of the REST-Q Fire to represent teamwork-related stressors and resources in firefighting was successful.

**Relationship of teamwork-related stressors and resources to stress.** Furthermore, correlations with stress were calculated to examine the newly developed items. This tested the assumption that the stressors measured by the REST-Q Fire are positively related to the general

stress perception during firefighting operations. The resources, on the other hand, are negatively related to this stress perception. When demonstrated, it confirms the content alignment of the newly developed items.

In the same survey as described above, respondents were also asked about their general experience of stress during a firefighting operation ("What is your average stress level in firefighting operations?"). The self-created item was measured on a ten-point Likert scale (1 = very low to 10 = very high). Overall scores were created for the frequency of stressors, the frequency of resources, the intensity of stressors, and the intensity of resources, and these were correlated with stress. Outliers were removed based on 2.5 standard deviations and only participants who had completed at least two categories were considered.

The results showed no significant correlations between stress and the frequency of stressors, the frequency of resources, and the intensity of resources, whereas the intensity of stressors was positively related to stress ($r = .23$, $p < .001$) (see Table 2 for results).

To look further into the effects, the intensity of stressors was divided into the 6 superordinate categories described in the development of the REST-Q Fire and shown in Table 1. Cronbach's α was between .70 and .87 for the different scales. It could be shown that all categories individually correlate positively with stress. For the other superordinate categories and subcategories, no significant effects were shown, except for the intensity of Decision-making as a resource, which was positively related to stress (see Table 2 for results).

**Further relationships.** Furthermore, the REST-Q Fire has already been used in two studies, using the same dataset as the one presented here, in which it was related to different scales. A study by Hagemann et al. [51] investigated the relationship between risky decision-making, conflicting goals, the frequency of teamwork-related stressors in firefighting, and debriefing as a countermeasure. Moreover, Aust et al. [52] studied the relation between teamwork-related stressors and resources and team flow. Their results will be summarized in the following.

In the study by Hagemann et al. [51], conflicting goals were measured with two self-created items focusing on the tension between (1) self-protection and achievement of the operational objective ("In firefighting operations, I experience conflicts between self-protection and achieving the operational objective.") and (2) acting fast vs. acting safely ("In firefighting operations, I experience conflicts between acting fast and acting safely."). The items were rated on a five-point Likert scale (1 = never to 5 = very often or always).

Results showed that conflicting goals were associated with risky decision-making and unsafe behavior. The frequency of teamwork-related stressors (communication, leadership, and shared mental models) mediated the relationship between debriefings and conflicting goals inasmuch as debriefings led to less frequent teamwork-related stressors, which in turn led to fewer conflicting goals [51].

Aust et al. [52] showed that the frequent experience of teamwork-related stressors is associated with less frequent team flow, which can be described as "a shared experience of flow during the execution of interdependent personal tasks in the interest of the team, originating from an optimized team dynamic" [53, p. 28]. The more often team-relevant resources are experienced, the more often team flow occurs. In an exploratory approach, it was found that the stressors regarding shared mental models, leadership, organization/coordination, and supportive behavior were significantly negatively associated with team flow. In contrast, for the resources, the subcategories communication, shared mental models, organization/coordination, and supportive behavior were significant positive predictors [52].

**Summary.** Overall, the assumed structure of the REST-Q Fire was confirmed. Furthermore, the intensity of stressors as a total value, as well as all six superordinate categories, correlated positively with stress. For the frequency of stressors, it was shown that three of the six superordinate categories (communication, leadership, and shared mental model) were

positively related to conflicting goals. So the use of debriefings could reduce the frequency of the stressors and thus also of the conflicting goals. Additionally, the frequency of stressors was negatively associated with team flow, while the frequency of resources was positively associated with team flow.

## Analysis of frequencies and intensities of teamwork-related stressors and resources

The aim of this study was the development of the measurement tool REST-Q Fire to quantify which teamwork-related stressors and resources are relevant to the field of firefighting and how regularly and intensively firefighters are confronted with these. Thus, the collected data was analyzed descriptively using SPSS Version 28 and the results are presented systematically in the next section.

## Results regarding frequencies and intensities of teamwork-related stressors and resources

In general, it was discovered that the resources are perceived more frequently and intensively than the stressors. The most frequent stressors were *task allocation*, *amount of information*, and *behavior of leaders*, while the most intense ones were *standards and safety measures*, *reliability of team members*, and *structure/ hierarchy*. The two most frequent stressors *task allocation* and *amount of information* are perceived as less stressful compared to the other stressors. The concordance between frequency and intensity was rather small. All items that capture stressors are shown in Table 3. The mean frequency represents how often the respective stressor is experienced on average during firefighting operations. The rank frequency was assigned according to the mean frequency. The stressor with the highest frequency is ranked 1, the stressor with the second highest frequency is ranked 2, and the one with the lowest frequency is ranked 26. The same procedure was followed for intensity. The mean intensity indicates how stressful firefighters experience the respective stressor. The stressor with the highest intensity is ranked 1 and so on to the stressor with the lowest intensity, which is ranked 26. Table 3 is sorted by descending frequency of occurrence. Thus, the first row shows the stressor that was mentioned most frequently, followed by the second most frequent stressor, and so on. The rated intensity of the stressors is also shown in this table.

The most frequent resources were *standards and safety measures*, *reliability of team members*, and *knowledge about skills and behavior of team members*. Unlike the stressors, the results here showed that the intensity displayed a quite similar result with *standards and safety measures*, *knowledge about skills and behavior of team members*, and *watch out for/ check on each other* as the most intense resources. Thus, it follows that the areas of *knowledge about skills and behavior of team members*, *watch out for/ check on each other*, and *reliability of team members* are experienced very frequently as well as perceived as very supportive. In general, even the less frequent resources are still frequent, and the intensity of support is above the midpoint of the scale for all resources. All items that capture resources are shown in Table 4. The mean frequency represents how often the respective resource is experienced on average during firefighting operations. The rank frequency was assigned according to the mean frequency. The resource with the highest frequency is ranked 1, the resource with the second highest frequency is ranked 2, and the one with the lowest frequency is ranked 23. The same procedure was followed for intensity. The mean intensity indicates how supportive firefighters experience the respective resource. The resource with the highest intensity is ranked 1 and so on to the resource with the lowest intensity, which is ranked 23. Table 4 is sorted by descending frequency of occurrence. Thus, the first row again shows the resource that was mentioned most

**Table 3. Stressor frequency and intensity sorted by descending frequency of occurrence.**

| Item | Mean frequency (SD) | n | Rank frequency | Rank intensity | Mean intensity (SD) | n |
|---|---|---|---|---|---|---|
| Task allocation | 2.91 (0.86) | 467 | 1 | 19 | 3.17 (0.99) | 443 |
| Amount of information | 2.85 (0.84) | 441 | 2 | 18 | 3.22 (1.09) | 423 |
| Behavior of leaders | 2.80 (0.83) | 450 | 3 | 5 | 3.59 (1.12) | 433 |
| Behavior of team members | 2.77 (0.75) | 446 | 4 | 4 | 3.59 (1.05) | 436 |
| Decision in case of situation change/ decide between different alternatives 2 | 2.69 (0.95) | 449 | 5 | 20 | 3.16 (1.10) | 407 |
| Hierarchy/ followership | 2.59 (0.98) | 445 | 6 | 23 | 3.07 (1.15) | 383 |
| Self-initiative and independence | 2.57 (0.84) | 461 | 7 | 6 | 3.56 (1.13) | 426 |
| Knowledge about common course of action | 2.55 (0.83) | 452 | 8 | 21 | 3.15 (1.02) | 413 |
| Quality of information | 2.52 (0.75) | 444 | 9 | 17 | 3.24 (1.02) | 418 |
| Self-initiative and independence 2 | 2.52 (0.83) | 461 | 10 | 15 | 3.27 (1.09) | 418 |
| Structure/ hierarchy | 2.52 (0.93) | 453 | 11 | 3 | 3.64 (1.10) | 395 |
| Communication with third parties | 2.41 (0.82) | 441 | 12 | 26 | 2.95 (1.08) | 388 |
| Supportive behavior/ support not possible | 2.35 (0.87) | 439 | 13 | 22 | 3.14 (1.06) | 375 |
| Insufficient consideration of hazards and operational standards | 2.35 (0.82) | 448 | 14 | 8 | 3.48 (1.08) | 392 |
| Passing on information between the fire brigade's task forces | 2.35 (0.79) | 447 | 15 | 9 | 3.46 (1.02) | 395 |
| Reliability of team members | 2.32 (0.74) | 434 | 16 | 2 | 3.64 (1.05) | 390 |
| Passing on information within the squad | 2.32 (0.81) | 450 | 17 | 11 | 3.45 (1.02) | 385 |
| Knowledge about skills and behavior of team members | 2.31 (0.83) | 445 | 18 | 24 | 3.05 (1.05) | 382 |
| (Support of) Decision makers | 2.30 (0.86) | 455 | 19 | 12 | 3.37 (1.07) | 371 |
| Watch out for/ check on each other | 2.15 (0.78) | 441 | 20 | 7 | 3.56 (1.03) | 360 |
| Interpositional knowledge | 2.09 (0.91) | 442 | 21 | 25 | 3.01 (1.10) | 323 |
| Decision in case of situation change/ decide between different alternatives | 2.06 (0.75) | 447 | 22 | 13 | 3.29 (1.05) | 348 |
| Shared Situational Awareness | 2.01 (0.84) | 430 | 23 | 10 | 3.45 (1.09) | 302 |
| Standards and safety measures | 2.00 (0.75) | 447 | 24 | 1 | 4.06 (1.05) | 341 |
| Supportive behavior/ support not possible 2 | 1.95 (0.85) | 441 | 25 | 16 | 3.27 (1.14) | 302 |
| Respond to the needs of others | 1.78 (0.79) | 418 | 26 | 14 | 3.28 (1.12) | 246 |

frequently, followed by the second most frequent resource, and so on. The rated intensity of the resources is also shown in this table.

## Discussion

The present study aimed to develop a measurement tool to evaluate the frequency and intensity of teamwork-related stressors and resources in firefighting operations. This tool should be able to identify the need for action in terms of interventions to promote teamwork. To this end, the REST-Q Fire was developed from qualitative research. CFAs confirmed the superordinate categories and subcategories identified in the qualitative research. The reported studies demonstrated the association of teamwork-related stressors and resources with stress, conflicting goals, and team flow. The results of the online survey using the newly developed tool showed the most frequent and intense stressors and resources.

Initially, the research gap was described that research on occupational stressors in firefighting operations mainly deals with organizational and operational stressors and not with teamwork-related stressors. As shown above, the present study contributes to closing this research gap. In addition, there is now a questionnaire, the REST-Q Fire, to gather teamwork-related resources in firefighting operations in a standardized way. By combining the frequency and

**Table 4. Resource frequency and intensity sorted by descending frequency of occurrence.**

| Item | Mean frequency (SD) | n | Rank frequency | Rank intensity | Mean intensity (SD) | n |
|---|---|---|---|---|---|---|
| Standards and safety measures | 4.04 (0.65) | 456 | 1 | 1 | 4.27 (0.87) | 453 |
| Reliability of team members | 3.96 (0.51) | 440 | 2 | 7 | 4.16 (0.73) | 440 |
| Knowledge about skills and behavior of team members | 3.96 (0.63) | 452 | 3 | 2 | 4.24 (0.78) | 452 |
| Watch out for/ check on each other | 3.96 (0.70) | 449 | 4 | 3 | 4.20 (0.80) | 449 |
| (Support of) Decision makers | 3.92 (0.70) | 459 | 5 | 14 | 4.10 (0.81) | 457 |
| Interpositional knowledge | 3.91 (0.78) | 450 | 6 | 13 | 4.11 (0.86) | 446 |
| Structure/ hierarchy | 3.87 (0.76) | 455 | 7 | 6 | 4.17 (0.87) | 453 |
| Passing on information within the squad | 3.85 (0.72) | 455 | 8 | 15 | 4.09 (0.84) | 450 |
| Passing on information between the fire brigade's task forces | 3.84 (0.66) | 450 | 9 | 11 | 4.12 (0.83) | 448 |
| Supportive behavior | 3.79 (0.77) | 445 | 10 | 12 | 4.11 (0.78) | 440 |
| Quality of information | 3.76 (0.62) | 445 | 11 | 17 | 4.07 (0.80) | 443 |
| Shared Situational Awareness | 3.75 (0.92) | 440 | 12 | 16 | 4.08 (0.91) | 428 |
| Knowledge about common course of action | 3.75 (0.67) | 456 | 13 | 9 | 4.13 (0.72) | 453 |
| Respond to the needs of others | 3.72 (0.95) | 443 | 14 | 10 | 4.12 (0.82) | 434 |
| Behavior of leaders | 3.72 (0.64) | 451 | 15 | 4 | 4.20 (0.82) | 450 |
| Consideration of hazards and operational standards | 3.68 (0.84) | 450 | 16 | 5 | 4.18 (0.83) | 440 |
| Behavior of team members | 3.58 (0.67) | 447 | 17 | 8 | 4.15 (0.79) | 446 |
| Amount of information | 3.56 (0.72) | 443 | 18 | 18 | 3.97 (0.86) | 441 |
| Communication with third parties | 3.55 (0.85) | 444 | 19 | 19 | 3.85 (0.86) | 432 |
| Task allocation | 3.50 (0.79) | 466 | 20 | 20 | 3.81 (0.89) | 463 |
| Self-initiative and independence | 3.46 (0.73) | 464 | 21 | 21 | 3.76 (1.03) | 460 |
| Decision in case of situation change/ decide between different alternatives | 3.37 (0.82) | 454 | 22 | 23 | 3.69 (0.86) | 445 |
| Hierarchy/ followership | 3.28 (0.83) | 454 | 23 | 22 | 3.70 (0.99) | 450 |

It is noticeable that the concordance between frequency and intensity is higher for resources than for stressors. With the latter, there tend to be more items that are either frequent or intense.

intensity of stressors and resources, it is possible to prioritize them and use these results for further implications.

## Results of Step 1

The development of the category system in Step 1 showed that, overall, the usual non-technical skills, well-known in the literature (see e.g. [2, 35, 38–40, 54]), also seem to be relevant for firefighting operations. In the literature, the category *Supportive behavior* is not as common as *Communication* or *Decision-making*. However, in our interviews and document analysis it proved to be an essential subject. Supportive behavior could be particularly important in firefighting operations in comparison to other HRTs because firefighters risk their own lives. Good collaboration is also very important for teams in health care, where much of the non-technical skills research comes from. Healthcare teams, though, do not risk their health or lives if something goes wrong.

In general, the items developed in Step 1 through qualitative research proved to be relevant for firefighting operations since most stressors and all resources have been experienced by each participant of the online questionnaire at least once. Likewise, the developed superordinate categories and subcategories of the REST-Q Fire were mostly confirmed and can thus be used for further research. It should be kept in mind that each individual item also has a specific content value and is thus very well suited to indicate specific training needs.

### Results of the frequencies and intensities in Step 2

In Step 2, the application of the REST-Q Fire showed that important stressors could be found in the areas of *behavior of leaders* and *behavior of team members*. Hectic, uncertain, and unco- ordinated behavior of both leaders and team members is therefore often experienced in fire- fighting operations and causes great stress. In the interviews, it was mentioned that when team members are hectic or stressed, the stress or rush spills over to others. This phenomenon has already been confirmed in studies (see e.g. [55]). For teamwork, this means that the stress of one single team member can spread, in the worst case, leading to the entire team no longer being able to work in a concentrated manner. This shows all the more how important stress prevention is for each team member. Demonstrating ways a firefighter can deal with his or her stress as well as with stressed team members is a training need identified here. If, for example, they manage to remain calm and maintain or build up the operational structure, this can in turn have a positive effect on the others.

In terms of resources, the areas *knowledge about skills and behavior of team members*, *watch out for/ check on each other*, and *reliability of team members* showed to be particularly valuable. On the one hand, these are experienced frequently and, on the other hand, also perceived as very supportive. This shows the importance of shared mental models. These depend, among other things, on the team members being aware of the prior knowledge and skills of other team members to better understand what to expect from them [41, 43]. Debriefings after oper- ations or operational exercises are a central measure here. They can intensively contribute to the team members getting to know each other (better) professionally and thus strengthen their shared mental model [51, 56, 57]. In the interviews, it was also mentioned that it is important for successful teamwork to know one's teammates to be able to rely on them. This includes the occasional exchange outside the job on personal topics. Many studies show similar findings, e.g. that trust is vital in teamwork [3, 37, 38].

The overview of the important teamwork-related stressors and resources shows that all are very closely linked to the relationship with the team members (their behavior, reliability, mutual support), which underlines the importance of interpersonal aspects, such as knowing each other or trust. This seems logical because many other outcomes depend on exactly these factors. For example, communication or coordination cannot work well if not everyone in the team is pulling together. Marks et al. [58] have also already addressed this aspect, citing inter- personal processes as a relevant factor in every phase of teamwork.

All things considered, the frequent and intense stressors and the frequent and supportive resources are important training needs, especially for trainees in the fire service. They can thus be given the basics crucial for teamwork in firefighting operations.

In detail, it is noticeable that the concordance between frequency and intensity is higher for resources than for stressors. With the latter, there tend to be more items that are either fre- quent or intense. In contrast to this, it is much clearer to classify which resources are particu- larly important and which are less important. However, even the less frequent resources are still frequent, and the intensity of support is above the midpoint of the scale for all resources. This suggests that the resources may already be better developed than the stressors and that the additional need for action is smaller here. Some stressors occur very rarely. Nevertheless, these should not be omitted in the further use of the REST-Q Fire, since they all have at least a medium intensity.

Furthermore, the results regarding *Standards and safety measures* are particularly striking. Adherence to these occurs frequently and is then perceived as very supportive. Violation of the standards and safety measures is not experienced very often but then leads to great stress. This means that this aspect is not only important for trainees in the fire service but could be

especially crucial for training more experienced firefighters. Consequently, awareness should be renewed that compliance with standards and safety measures is an invaluable resource.

Looking back on the previous research on occupational stressors, the five most common teamwork-related stressors from this study have all been described in the same or similar form. The categories are sometimes named slightly differently respectively are usually not sub-divided in as much detail in existing research as in the present study. Therefore, the results cannot always be assigned as they stand. Nevertheless, there is a high degree of agreement between the present results and the existing research literature.

The most common teamwork-related stressor in the present study is *task allocation*, which includes *conflicting tasks* described in existing research [13]. *Amount of information* (Rank 2) was used to cover either too much or too little information in the present study. This means that one characteristic corresponds to *lack of communication* [9]. The *behavior of leaders* (Rank 3), together with *structure/hierarchy* (Rank 11), creates an *inconsistent leadership style* [11]. *Behavior of team members* (Rank 4) is assigned to the superordinate category *support between colleagues* in the present study and can thus be linked to a *lack of support between colleagues* [11, 13]. *Decision-making*, which is mentioned frequently in other studies [9, 10, 12, 15, 25], is the superordinate category of *Decision in case of situation change/ decide between different alternatives 2* (Rank 5). The other aspects of the existing research *breach of standards* and *lack of situational awareness* [9, 10] are also reflected in the teamwork-related stressors of this study and thus show that the present results can systematically confirm and expand the beginnings that were already researched in the existing literature.

## Results of the correlations in Step 2

In Step 2, as a validation for the REST-Q Fire, the frequency and intensity of the teamwork-related stressors and resources were correlated with stress. The results showed that the intensity of the stressors correlates positively with stress. The frequency of the stressors however does not have a significant correlation with stress. Stress was queried similarly to the intensity of the stressors, namely with a scale measuring the level of stress. This could explain why there is a correlation with the intensities but not with the frequencies of the stressors. In contrast, a correlation with the frequencies of the stressors and resources is shown with team flow, which fits the above explanation, since it is also asked using a frequency scale. The intensity of resources as a total score as well as all the subcategories did not correlate with stress (except for decision-making, which has a small positive correlation with stress). There appear to be many other factors besides the intensity of resources that influence stress, such as perhaps the operational stressors mentioned in the introduction.

## Strengths and limitations

One strength of this study is the mixed methods approach used to develop the REST-Q Fire. Furthermore, a sufficiently large and very specific sample was acquired. Accordingly, the results are very transferable to the fire department context. In addition, the development of the REST-Q Fire lays the foundation for further research that can specifically address teamwork-related stressors and resources in the fire service. On the other hand, the first limitation of the study might be that the online questionnaire was very long and many participants did not complete it fully. It is highly likely that only the very motivated people stuck with it and completed the questionnaire to the end. This means that there is probably also a bias in this respect, that mainly people who are very interested in the topic completed the questionnaire. The motivation of the participants could possibly distort the results to the effect that a high motivation to participate arises when firefighters have a lot to report—either because they are very satisfied

with the course of the firefighting operations or because they are very dissatisfied with the course of the firefighting operations. The present results could therefore over-represent the extremes. However, since we countered non-completion of the questionnaire by randomizing the items, we were still able to achieve a high and roughly equal number of participants for each item. Secondly, the different years of experience might also have an impact on the assessment of stressors and resources–both in the interviews and in the REST-Q Fire. Specifically, the stressors show small positive correlations with the years of experience and the number of firefighting operations in the sample for the REST-Q Fire. Additionally, the survey was completed retrospectively, and the firefighters were asked to average the frequency and intensity of stressors and resources across all experienced firefighting operations. Especially in cases of long professional experience or many performed operations, biases might have occurred in the sense that a concise firefighting operation was particularly remembered. Furthermore, the sample consists of 94.5% men and only 5.5% women. However, this is representative of the firefighting sector in Germany, where women accounted for 2.3% of professional firefighters, 10.5% of volunteer firefighters, and 3.7% of industrial firefighters in 2020 [59]. Nevertheless, future studies should focus more on women and what they experience in firefighting operations to determine whether their experiences differ significantly from those of men. Thirdly, the construction of the REST-Q Fire was not executed classically with the help of statistic item characteristics and based on a large item pool that was gradually reduced. Much more emphasis was placed on the experience of the experts and situations from the interviews were integrated into the REST-Q Fire. No items were subsequently removed since all generated items were relevant and an exclusion would have led to a loss of information. Generally, an evaluation at the level of individual items is recommended since the very specifically formulated items can provide precise information about the needs of a team. The CFAs conducted afterward with the collected data confirmed the assumed structure to a large extent. However, it should be noted that the internal consistencies are not always sufficient. If necessary, the subcategories can also be selected, but the REST-Q Fire is rather indented to be a measurement tool for single facets in a practical context. The development of the tool was structured for this purpose and because of the extensive questionnaire, it was not possible to include more than one item per facet.

## Implications

The REST-Q Fire was developed to identify relevant stressors and resources in a firefighting operation to design a training course for trainees in the fire service. Furthermore, it is intended for use in a practical context. Thanks to the REST-Q Fire, teams can determine areas of teamwork still requiring training and select specific training content. For example, after a firefighting operation, the REST-Q Fire can be completed concerning exactly that one firefighting operation. The teamwork-related resources and stressors in this firefighting operation are then made visible. This involves evaluating which are the most frequent and which are the most intensive teamwork-related stressors and resources. Training can then be tailored to precisely these. This means that the non-technical skills that appeared to be particularly relevant in the firefighting operation and were not solved well can be trained. Once training has been carried out, the questionnaire can be used again after firefighting operations to see whether the training has had an effect and whether the values on the corresponding scales have changed positively. If the REST-Q Fire is completed regularly after firefighting operations, it is possible to identify where the greatest need for action exists with regard to teamwork competencies. Since non-technical skills, such as those described here in the teamwork-related stressors and resources, are an important aspect of successful team performance [24, 35, 36], training them

is a promising approach to improving team performance (e.g. [60]). However, it is necessary to analyze needs before developing or delivering training (e.g. [54]), so that the content is targeted to the audience and resources are used optimally. This is comparable to the SIRE, a questionnaire for emergency responders to measure stress in rescue service missions [61]. The REST-Q Fire can also be used for trainees in the fire service to show how perceived teamwork-related stressors and resources vary during an apprenticeship in the fire service. From this, specific training needs can be derived for different phases of the apprenticeship. Furthermore, the REST-Q Fire can be used to evaluate training success by using it before and after the training and comparing the results.

In summary, the need for the present study arises more from the literature. The literature shows that teamwork is responsible for errors and critical situations. It also shows that teamwork can be trained, while many other stressors cannot be influenced. Ultimately, however, occupational stressors are a larger package that needs to be managed. For a complete reduction of stress and a comprehensive increase in occupational safety, for example, managerial stressors must also be addressed to reduce the general stress level of firefighters outside of operations.

Regarding future research, the REST-Q Fire can be used to advance research in teamwork in the firefighting context. Future studies could, for example, focus not only on a retrospective survey but also on using the REST-Q Fire directly after firefighting operations or operational exercises to collect more specific data on the frequency and intensity of teamwork-related stressors and resources. Moreover, the data collected in this way can be compared with observations from operational exercises to have an objective view of the frequency of teamwork-related stressors and resources. The intensity data can be compared with physiological stress markers recorded in the operational exercises.

Furthermore, the influence of teamwork-related stressors and resources on various outcome measures, such as performance or physiological stress measures, could be tested. Additionally, there should be a future focus on the provision of normative values so that firefighters can be informed about whether the level of stressors and resources in their teams is of concern. Likewise, it would be interesting to be able to use the questionnaire not only for firefighting but also to adapt it for other contexts of firefighting work, such as technical assistance, and discover if the teamwork-related stressors and resources are similar in that area.

## Conclusion

In summary, the present study contributes to the research on teamwork-related stressors and resources in firefighting operations by exploring the frequency and intensity of these and relating them to other constructs. This has resulted in the REST-Q Fire, a questionnaire that has multiple uses in firefighting education and training evaluation.

## Supporting information

**S1 Table. Teamwork-related stressors and resources during firefighting operations.** (DOCX)

## Author Contributions

**Conceptualization:** Lena Heinemann, Fabienne Aust, Corinna Peifer, Maik Holtz, Vera Hagemann.

**Data curation:** Lena Heinemann, Fabienne Aust.

**Formal analysis:** Lena Heinemann, Fabienne Aust.

**Funding acquisition:** Corinna Peifer, Maik Holtz, Vera Hagemann.

**Investigation:** Lena Heinemann, Fabienne Aust, Maik Holtz.

**Methodology:** Lena Heinemann, Fabienne Aust, Corinna Peifer, Vera Hagemann.

**Project administration:** Corinna Peifer, Maik Holtz, Vera Hagemann.

**Resources:** Corinna Peifer, Maik Holtz, Christian Miller, Vera Hagemann.

**Supervision:** Corinna Peifer, Vera Hagemann.

**Visualization:** Lena Heinemann.

**Writing – original draft:** Lena Heinemann, Fabienne Aust.

**Writing – review & editing:** Corinna Peifer, Maik Holtz, Christian Miller, Vera Hagemann.

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
