## [Decision Letter · Decision Letter 0]

14 Mar 2023

PONE-D-23-00472Development and application of a questionnaire, the REST-Q, to identify teamwork-related resources and stressors in firefighting operationsPLOS ONE

Dear Dr. Heinemann,

Thank you for submitting your manuscript to PLOS ONE. After careful consideration, we feel that it has merit but does not fully meet PLOS ONE’s publication criteria as it currently stands. Therefore, we invite you to submit a revised version of the manuscript that addresses the points raised during the review process.

We look forward to receiving your revised manuscript.

Kind regards,

Boshra Ismael Ahmed Arnout

Academic Editor

PLOS ONE

Journal Requirements:

“This work was supported by the German Social Accident Insurance (DGUV) [grant number FP-433]. The funders had no role in study design, data collection and analysis, decision to publish, or preparation of the manuscript.”

Additional Editor Comments (if provided):

Dear Author

The paper PONE-D-23-00472 has been reviewed by experts in the field who consider that the paper can publish after major revision. For your guidance, you can benefit from the reviewer's comments are appended below.

We wish you a meaningful day.

Yours Sincerely

Reviewers' comments:

Reviewer's Responses to Questions

**Comments to the Author**

1. Is the manuscript technically sound, and do the data support the conclusions?

Reviewer #1: Yes

Reviewer #2: Partly

Reviewer #3: Yes

2. Has the statistical analysis been performed appropriately and rigorously? 

Reviewer #1: Yes

Reviewer #2: Yes

Reviewer #3: Yes

3. Have the authors made all data underlying the findings in their manuscript fully available?

Reviewer #1: No

Reviewer #2: Yes

Reviewer #3: Yes

4. Is the manuscript presented in an intelligible fashion and written in standard English?

Reviewer #1: Yes

Reviewer #2: Yes

Reviewer #3: Yes

5. Review Comments to the Author

Reviewer #1: - Revise the title to reflect objects of the manuscript, Suggestion title: Development and Validation of the REST-Q in firefighting operations.

-REST-Q can be RSTF-Q.

-Abstract conclusion is recommended to include something about: sample mean, standard deviation and Statistical results.

- It is best to do an exploratory factor analysis (EFA) first before confirmatory factor analysis (CFA).

-All the participants of this study approximately are male. This need to be included in the limitation and could add recommendation for including more female in the future.

Reviewer #2: The manuscript described a technically sound piece of scientific research about :Development and application of a questionnaire, the REST-Q, to identify teamwork-related resources and stressors in firefighting operations

with data that supports the conclusions. Experiments have been conducted rigorously, with appropriate controls, replication, and sample sizes. The conclusions had been drawn appropriately based on the data presented.

The research topic , please see attachement.

Reviewer #3: This manuscript aims to develop a measurement tool to identify teamwork-related resources and stressors in firefighting operations. The authors interviewed firefighters, then did a document analysis to indicate an overview of the teamwork stressors and resources. Next, they developed the REST-Q questionnaires and conducted them with 491 firefighters. A study like this is important, given the public health implications of the duty-related stresses in fire service personnel.

The title and abstract are appropriate, clear, and informative. Research problems are clear and concise. In the introduction section, the authors provided the rationale and the urgency of tool development.

The method section is well-described. The sample size and participants are clearly explained and serve the purpose of this study. However, I would like the authors to clarify participant selection's inclusion and exclusion criteria. For example, in the interview process (Page 6, line 138), the authors stated, "experienced professional, volunteer and industrial firefighters." Please define experienced. Would anyone with work experience higher than three years be included in the study since among those who participated in the interview, their years of work experience varied from 3 to 35 years, regardless of their profession (Page 7, lines 158-167)? Would the gap of years of experience affect their points of view on firefighting operations? Similarly, in the review of the REST-Q, 491 firefighters completed the questionnaire, and they were different in demographics and years of working experience. For example, participants were approximately 95% male and 5% female and had experienced between 1 and 7500 firefighting operations. Would these variables, such as gender or firefighting operations, affect the outcomes of the tools? And this should be included and addressed in the discussion section.

The authors stated on Page 11, lines 256-257 that only those participants who completed the respective category in full were used in the CFA. Did 491 participants complete all categories, and were all 491 included in the CFA? Please clarify. Also, please specify the N in Tables 2, 3, and 4. I would also like the authors to explain and define rank frequency and intensity. How do the authors utilize these terms if the rank numbers are different? What is the underly meaning of the mean frequency and mean intensity? They only explained the results from Tables 3 and 4 without further details (Pages 16 and 17).

Some content in the discussion section should move to the result section. Since the study is more into tools development, the authors clearly explained and discussed their findings. Overall, this is a nicely written paper presenting valuable data on information regarding tools that can be used to identify teamwork-related resources and stressors in firefighting operations. It will significantly contribute to the literature in these critical public health and fire service areas.

6. PLOS authors have the option to publish the peer review history of their article (what does this mean?). If published, this will include your full peer review and any attached files.

Reviewer #1: No

Reviewer #2: **Yes: **Dr. Mohamed A. Gharib

Reviewer #3: No

---

## [Author Response · Author response to Decision Letter 0]

6 Apr 2023

General

R1: Revise the title to reflect objects of the manuscript, Suggestion title: Development and Validation of the REST-Q in firefighting operations. - Thank you for the idea. We integrated validation into the title. Since we find the aspect of non-technical skills essential for this manuscript, we would like to keep this part of the title.

R1: REST-Q can be RSTF-Q. - Thank you for your suggestion. We decided for REST-Q since it sounds similar to ‘rescue’ and therefore reflects the content of the questionnaire.

R2: The manuscript described a technically sound piece of scientific research about: Development and application of a questionnaire, the REST-Q, to identify teamwork-related resources and stressors in firefighting operations with data that supports the conclusions. Experiments have been conducted rigorously, with appropriate controls, replication, and sample sizes. The conclusions had been drawn appropriately based on the data presented. The research topic , please see attachement. - Thank you for your friendly comment. Unfortunately, we were not able to find any attachement. It would be great if you could please specify your comment regarding the research topic.

R3: This manuscript aims to develop a measurement tool to identify teamwork-related resources and stressors in firefighting operations. The authors interviewed firefighters, then did a document analysis to indicate an overview of the teamwork stressors and resources. Next, they developed the REST-Q questionnaires and conducted them with 491 firefighters. A study like this is important, given the public health implications of the duty-related stresses in fire service personnel. - We appreciate your kind assessment. Thank you.

R3: Overall, this is a nicely written paper presenting valuable data on information regarding tools that can be used to identify teamwork-related resources and stressors in firefighting operations. It will significantly contribute to the literature in these critical public health and fire service areas. - Thank you for your friendly comment. We are happy to hear that you value the usefulness of the contribution.

R1: Abstract conclusion is recommended to include something about: sample mean, standard deviation and Statistical results. - We appreciate your recommendation and included statistical results in the abstract (Page 1, lines 26-32).

R3: The title and abstract are appropriate, clear, and informative. Research problems are clear and concise. - Thank you.

R3: In the introduction section, the authors provided the rationale and the urgency of tool development. - Thank you.

R1: It is best to do an exploratory factor analysis (EFA) first before confirmatory factor analysis (CFA). - We appreciate your thoughtful feedback. This is a point we also discussed in the team and decided to omit the exploratory factor analysis. Since we developed the questionnaire based on our interviews and document analysis, we used the categories identified there. Thus, it was not our intention to examine the data in an exploratory way, but rather to use confirmatory factor analysis to check the structure of the data that we had assumed. Nevertheless, we conducted an exploratory factor analysis, with the result that only one meaningful factor emerged, since the teamwork aspects are closely related. However, from a theoretical point of view, they are distinguishable and can be promoted and trained individually. Therefore, it makes sense for us to still separate them for a practical tool like this.

R3: The method section is well-described. The sample size and participants are clearly explained and serve the purpose of this study. However, I would like the authors to clarify participant selection's inclusion and exclusion criteria. For example, in the interview process (Page 6, line 138), the authors stated, "experienced professional, volunteer and industrial firefighters." Please define experienced. Would anyone with work experience higher than three years be included in the study since among those who participated in the interview, their years of work experience varied from 3 to 35 years, regardless of their profession (Page 7, lines 158-167)? Would the gap of years of experience affect their points of view on firefighting operations? Similarly, in the review of the REST-Q, 491 firefighters completed the questionnaire, and they were different in demographics and years of working experience. For example, participants were approximately 95% male and 5% female and had experienced between 1 and 7500 firefighting operations. Would these variables, such as gender or firefighting operations, affect the outcomes of the tools? And this should be included and addressed in the discussion section. - Thank you for your constructive remark. We described the selection criteria more precisely by adding the criteria for experience in firefighting in order to be included in the study (Page 6, lines 137-138).

The further points were included in the limitations where we added more information regarding differences between gender and working experience (Page 22, lines 484-496).

R3: The authors stated on Page 11, lines 256-257 that only those participants who completed the respective category in full were used in the CFA. Did 491 participants complete all categories, and were all 491 included in the CFA? Please clarify. - Thank you for your thoughtful question. Not all participants completed all categories. We added the sample size for each category (Page 11, lines 258-263).

R3: Also, please specify the N in Tables 2, 3, and 4. - Thank you for the hint. We added the N in Tables 2, 3, and 4.

R3: I would also like the authors to explain and define rank frequency and intensity. How do the authors utilize these terms if the rank numbers are different? What is the underly meaning of the mean frequency and mean intensity? They only explained the results from Tables 3 and 4 without further details (Pages 16 and 17). - We appreciate your questions and added further explanation (Page 16, lines 367-374 & page 18, lines 387-393). The ranking lists for frequency and intensity were used separately. We used this approach to find stressors and resources which are at the same time frequent and intense, but we didn’t aggregate those rankings. 

R1: All the participants of this study approximately are male. This need to be included in the limitation and could add recommendation for including more female in the future. - Thank you for your constructive suggestion. We included this remark in the limitations (Page 23, lines 491-496).

R3: Some content in the discussion section should move to the result section. Since the study is more into tools development, the authors clearly explained and discussed their findings. - We appreciate your suggestion. We moved some content regarding frequency and intensity of stressors and resources to the results section (Page 16, lines 365-366; Page 18, lines 383-386 & Page 19, lines 398-399).

---

## [Decision Letter · Decision Letter 1]

5 Feb 2024

PONE-D-23-00472R1Development and validation of a questionnaire, the REST-Q, to identify teamwork-related resources and stressors in firefighting operationsPLOS ONE

Dear Dr. Heinemann,

Thank you for submitting your manuscript to PLOS ONE. After careful consideration, we feel that it has merit but does not fully meet PLOS ONE’s publication criteria as it currently stands. Therefore, we invite you to submit a revised version of the manuscript that addresses the points raised during the review process.

We look forward to receiving your revised manuscript.

Kind regards,

Roghieh Nooripour, Ph.D

Academic Editor

PLOS ONE

Reviewers' comments:

Reviewer's Responses to Questions

**Comments to the Author**

1. If the authors have adequately addressed your comments raised in a previous round of review and you feel that this manuscript is now acceptable for publication, you may indicate that here to bypass the “Comments to the Author” section, enter your conflict of interest statement in the “Confidential to Editor” section, and submit your "Accept" recommendation.

Reviewer #4: All comments have been addressed

2. Is the manuscript technically sound, and do the data support the conclusions?

Reviewer #4: No

3. Has the statistical analysis been performed appropriately and rigorously? 

Reviewer #4: No

4. Have the authors made all data underlying the findings in their manuscript fully available?

Reviewer #4: No

5. Is the manuscript presented in an intelligible fashion and written in standard English?

Reviewer #4: No

6. Review Comments to the Author

Reviewer #4: I recommend incorporating these suggestions to enhance clarity and overall effectiveness. Once the revisions are applied, I look forward to reviewing the updated version.

1. Strengthen transition sentences between paragraphs to ensure a seamless flow of ideas. For example, consider using transitional phrases to connect the discussion of environmental stressors to the introduction of teamwork skills.

2. Elaborate further on the challenges faced by firefighters beyond the statistical data. Providing real-life examples or scenarios can help readers relate more deeply to the issues discussed.

3. While FUK is explained upon its first mention, consider providing a brief reminder or explanation if the acronym is used later in the text to aid reader comprehension.

4. Ensure consistent use of terminology throughout the introduction. For instance, check that "teamwork stressors and resources" is consistently referred to by the same terminology.

5. Depending on the nature of your document, consider incorporating visual aids such as charts or graphs to illustrate the statistical data or complex concepts, enhancing reader understanding.

6. Introduce variety in sentence structure to maintain reader engagement. Mix shorter, impactful sentences with more complex ones for a balanced rhythm.

7. While the objectives are stated clearly, briefly clarify how the developed questionnaire will address the identified gaps and contribute to the existing body of knowledge.

8. Be mindful of word choice, opting for the most precise and impactful terms. This can contribute to a more polished and sophisticated writing style.

9. While detailed information is important, consider condensing certain sections for better readability. For example, the paragraph at the beginning of Step 2 could be concise.

10. Some sentences are lengthy and could be broken down to enhance clarity.

11. Explicitly mention the sampling strategy used for participant selection. This ensures transparency and helps readers understand the generalizability of your findings.

12. Include information about the response rate in the sample description to provide insights into potential biases.

13. Provide more details about the pretesting process, such as the feedback received, and how it informed modifications to the questionnaire.

14. When discussing the scales used for frequency and intensity, consider providing a brief rationale for choosing these specific scales.

15. Consider breaking down complex sentences into simpler ones for improved clarity.

16. Consider grouping related findings together to enhance readability.

17. While the strengths are well-highlighted, there could be a more detailed exploration of the limitations, particularly in terms of potential biases and their impact on the study's outcomes.

18. consider addressing potential alternative explanations or confounding stressors, resources, and other variables.

19. Expand on the suggestions for future research. Provide more specific recommendations for researchers who might want to build on this study.

20. Provide additional details on how exactly the REST-Q could be implemented in a training or evaluation setting. Offer examples or scenarios to illustrate its practical use.

7. PLOS authors have the option to publish the peer review history of their article (what does this mean?). If published, this will include your full peer review and any attached files.

Reviewer #4: No

---

## [Author Response · Author response to Decision Letter 1]

21 Mar 2024

We responded to the reviewers' comments in the document "Response to the reviewers."

---

## [Decision Letter · Decision Letter 2]

10 May 2024

Development and validation of a questionnaire, the REST-Q Fire, to identify teamwork-related resources and stressors in firefighting operations

PONE-D-23-00472R2

Dear Dr. Heinemann,

We’re pleased to inform you that your manuscript has been judged scientifically suitable for publication and will be formally accepted for publication once it meets all outstanding technical requirements.

Kind regards,

Steve Zimmerman, PhD

Senior Editor, PLOS ONE

Additional Editor Comments (optional):

Please note that reviewer 5 has some suggestions for improvements to your paper. I think that some of these suggestions are outside the scope of the manuscript (e.g., a description of content analysis; discussing a study on "predicting aggression among Iranian athletic adolescent girls"). If you would like to incorporate any of the reviewer's other suggestions into your final manuscript you may, but acceptance is not contingent on doing so.

Reviewers' comments:

Reviewer's Responses to Questions

**Comments to the Author**

1. If the authors have adequately addressed your comments raised in a previous round of review and you feel that this manuscript is now acceptable for publication, you may indicate that here to bypass the “Comments to the Author” section, enter your conflict of interest statement in the “Confidential to Editor” section, and submit your "Accept" recommendation.

Reviewer #4: All comments have been addressed

Reviewer #5: (No Response)

2. Is the manuscript technically sound, and do the data support the conclusions?

Reviewer #4: Yes

Reviewer #5: Yes

3. Has the statistical analysis been performed appropriately and rigorously? 

Reviewer #4: Yes

Reviewer #5: No

4. Have the authors made all data underlying the findings in their manuscript fully available?

Reviewer #4: Yes

Reviewer #5: No

5. Is the manuscript presented in an intelligible fashion and written in standard English?

Reviewer #4: Yes

Reviewer #5: Yes

6. Review Comments to the Author

Reviewer #4: I am pleased to inform you that the revisions to your manuscript have been meticulously incorporated, resulting in a thoroughly corrected and now deemed acceptable document. I extend my sincere appreciation for your committed efforts and collaborative approach, which have significantly enhanced the overall quality of your article.

Reviewer #5: 1- Strengthen the statement of the problem by referring to the following studies and emphasizing the existing research gap.

2- Revise the introduction and deal more coherently with your research problem and the current research literature.

3- On what basis are the interview questions designed?

4- On what basis was the sample size of the interviewees determined? Did you use the theoretical saturation method? Explain.

5- Describe the method of content analysis.

6- State the validity and reliability of your tool clearly.

7- What was the sampling method in the second stage? Provide the sampling formula.

8- State the software used.

9- With what psychometric methods was your tool approved?

10- In order to explain your results, refer to the possible explanations of previous studies.

11-Explain the obtained factor loading more clearly.

12-12- To strengthen your study, you can refer to the link below.

https://link.springer.com/article/10.1186/s40359-022-00852-2

7. PLOS authors have the option to publish the peer review history of their article (what does this mean?). If published, this will include your full peer review and any attached files.

Reviewer #4: **Yes: **Roghieh Nooripour

Reviewer #5: No

---

## [Editor Report · Acceptance letter]

17 May 2024

PONE-D-23-00472R2 

PLOS ONE

Dear Dr. Heinemann, 

I'm pleased to inform you that your manuscript has been deemed suitable for publication in PLOS ONE. Congratulations! Your manuscript is now being handed over to our production team.

Kind regards, 

on behalf of

Dr Steve Zimmerman 

Staff Editor

PLOS ONE